# Higher Levels of Early Childhood Caries (ECC) Is Associated with Developing Psychomotor Deficiency: The Cross- Sectional Bi-Township Analysis for The New Hypothesis

**DOI:** 10.3390/ijerph16173082

**Published:** 2019-08-24

**Authors:** Chen-Yi Liang, Yen-Chun G. Liu, Tien-Yu Shieh, Yi-Chun Tseng, Andy Yen-Tung Teng

**Affiliations:** 1Graduate Institute of Dental Sciences and School of Dentistry, College of Dental Medicine, Kaohsiung Medical University, Kaohsiung City 80708, Taiwan; 2Department of Childhood Education and Nursery, Chia Nan University of Pharmacy & Science, Tainan City 71710, Taiwan; 3Graduate Institute of Dental Sciences, Dept. of Oral Hygiene, College of Dental Medicine, Kaohsiung Medical University, Kaohsiung City 80708, Taiwan; 4Center for Osteoimmunology and Biotechnology Research (COBR), School of Dentistry, College of Dental Medicine, Kaohsiung Medical University KMU Hospital, Kaohsiung City 80708, Taiwan; 5Lab of Molecular Microbial Immunity, Div. of Periodontology, the Eastman Institute for Oral Health (EIOH), School of Medicine and Dentistry, University of Rochester, Rochester, NY 14620, USA

**Keywords:** severe childhood caries (ECC) and dmft scores, psychomotor vs. language development, CCDI vs. MCDI, pre-school children

## Abstract

The aim of this study was to reassess and confirm the relationship between early childhood caries (ECC) and manifestations of psychomotor deficiency in 4–6-yr-old kindergarteners, which has remained elusive to date. A cross-sectional study with bi-township analysis was designed whereby 353 kindergarteners, aged 4–6 whose caries were greater (dmft (decayed, missing and filled teeth, dmft index) = 5.25) than that of the national average, located in a rural township of central Taiwan were recruited using simple random-selection. Besides the personal, demographic, and dietary information, the measurements for caries and the amended comprehensive scales (CCDI) of children’s psychomotor development were used to address their relationship. One-way ANOVA vs. multiple linear regression were employed to compare the differences of variables between age, gender, BMI (Body Mass Index), and dmft scores vs. relationships among all variables, respectively. The results confirmed that there was a positive relationship between severe ECC (dmft > 3~8) and psychomotor deficiency (i.e., expressive language and comprehension-concept scales, etc.) amongst the kindergarteners analyzed. Our cross-sectional bi-township analysis has confirmed that there is indeed an association between severe ECC and psychomotor deficiency in kindergarteners, and we suggest that this may arise through critical stages of growth, not only via personal language communications, but psycho-social engagements as well. Therefore, a new hypothesis is proposed.

## 1. Introduction

Dental caries is one of the most common and preventable childhood diseases and continues to affect susceptible dentitions throughout adulthood in life [1]. The WHO defines dental caries as “a localized, post-eruptive, pathological process of external origin involving softening of the hard tooth structure and proceeding to the formation of a cavity”. Early childhood caries (or ECC) is the primary cause of oral pain and tooth loss, which can be arrested in its early stage, producing reduced tooth mineralization and clinically white-spot lesions [2,3]; however, it may not be self-limiting without proper care. 

Previous studies in the field have shown and suggested that high levels of ECC, or alike, may result in early decays [2,3], oral-facial pain, reduced dietary intakes, loss of weight or/and sleep, accompanied by a poorer quality of life [2,3,4,5,6]. These conditions could lead to lowered mastication on the occlusal forces, the brain’s neurologic activation, or even some interferences with the metabolism and intellectual development [7,8,9], where the appropriate stimuli are linked to the synapses on the glial (or dendritic) cells for the neuro-physiological functions [10,11]. It has been previously suggested that severe caries and ECC may affect children’s’ verbal skills for proper communications and also weight loss along with certain developmental delays or deficiencies [12,13]. To date, the physiological pathways linking the ECC and psychomotor deficiency remain rather elusive; however, it has been speculated that when caries activity is high (i.e., ECC), the children are prone to reduce chewing on foods and swallowing activities, resulting in digestive inefficiency, thus lowering their nutritional intake needed for growth [2]. In parallel, the resulting visual, auditory, proprioceptive or vestibular dysfunctions, if occurred, the affected sensory neurons may not function proficiently, leading to profound deficits in the development, learning, emotional maturity and stability, or/and social interactions of the host [14].

As ECC or severe caries may give rise to concerns with respect to general health, we have been interested in studying whether early caries (ECC) may be implicated in the overall psychomotor development of pre-school kindergarteners; in particular, its potential relationship to the personal-social development and/or expressive language manifestations, etc. [15], which remains unclear to date. To further reassess and confirm this intriguing issue, we randomly selected, along with the bi-township analysis for a parallel comparison, the aged 4–6 kindergarteners from a rural township of Nantou County in the central region of Taiwan, where caries (dmft (decayed, missing and filled teeth, dmft index) = 5.25) [16,17] was estimated much higher than those of the national surveys reported [18,19] and the southern cities of Taiwan which we reported recently (dmft = 4.07) [15]. Herein, we hypothesized to reassess whether the higher levels of ECC are indeed associated the child’s psychomotor deficiency or not (i.e., via the CCDI scale), by exploring any correlation that may arise from such differential caries activities measured through a bi-township analysis. Thereafter, a sequence of analyses, by dividing the caries scores (e.g., dmft) and the measurements of psychomotor development (e.g., CCDI), was employed and computed to address the proposed question of interest. Therefore, as the underlying variables or factors being analyzed for the new hypothesis proposed, the present findings, once fully substantiated, may contribute to our better understanding of its ultimate causes and/or associated risks, which will be applicable to developing new oral health-care protocols or novel strategies critical for the prevention of childhood dental diseases, like ECC, in the future.

## 2. Method and Materials 

The present cross-sectional study, with the bi-township analysis planned for a parallel comparison, was designed to reassess and confirm if any potential association(s) existed between higher levels of ECC and their physical or/and psychomotor deficiency in the kindergarteners of the rural township in the central region of Taiwan. Based on the previous surveys, the children’ caries activity in this region was estimated much higher than that of the national surveys (~4.35) [17] and that of the urban cities in southern Taiwan reported (dmft = 4.07) [15]. Herein, we employed the same measures and criteria [15], which are shown below regarding the detailed analyses for the objective as described above. 

### 2.1. Subject Selection

This study was first approved by the IRB and Ethics Committees of Kaohsiung Medical University, Kaohsiung, Taiwan (KMU-IRB#93-037). Informed consent was obtained after the protocol was thoroughly explained to the parents or legal guardians by the calibrated interviewers. We employed the published data of “National Dental Survey of Children under Six” [17], where our minimal sample size was estimated, *n* = 334, to reach the statistical significance, based on the criteria of the power analysis selected as follows: α = 0.05, SD = 5.5, power = 0.9 and a 30% study drop-out rate (as the rejection). Since we recently reported the study results derived from the urban cities located in southern Taiwan [15], after which the central region of Taiwan was then randomly selected out of the rest of the country’s geographic regions, including: the Northern, Eastern, Central and Mountain areas for the present cross-sectional study. Further, we included all registered 21 kindergartens as the total recruitment (*n* = 401), located in rural Xinyi township of the Nantou county, via the simple random sampling existed at the time, in the central region of Taiwan for the subsequent analyses to address the hypothesis. In the Nantou county, it was estimated to carry higher dmft activity (~5.25) [16] than that of the national average (~4.35) [17] and the southern cities that we recently reported (~4.07) [15]. Later, 353 aged 4–to-6-yr old kindergarteners completed the oral examinations (e.g., participation rate = 88.03%), where their gender, age, parental education and occupations, dmft, BMI (Body Mass Index), dietary and surveys of psychomotor development by the parents were collected and then all were entered into the present study for analyses. To our best knowledge, neither kindergarteners with special needs or physical disabilities, nor any obvious systemic disorder, were recruited as study subjects, consistent with the observation from our previous study in urban cities of the southern Taiwan, thereby suggesting that these two cohorts may carry comparable oral vs. general health characteristics, except the caries rates [15]. 

### 2.2. Caries Examination for the Dmft Score

All clinical examinations were performed based on the WHO guidelines (WHO 1997; oral health surveys: basic methods 4th Edition). Each participant received the oral examinations by two calibrated dentists, whose resulting diagnostic reproducibility was estimated κ = 0.80 – 0.84, using the results of 12 randomly selected kindergarten children, based on the same WHO guidelines. The oral examinations included the number of decayed, extracted, and filled primary teeth, where the sums of these measurements were computed for the dmft scores, accordingly.

### 2.3. Body Mass Index (BMI) Determination

BMI is a measure of weight by height that the WHO recommends, commonly used to classify the weight and obesity. It is defined as the weight in kg divided by the square of the height in meters (kg/m^2^). The normal BMI for pre-school children was defined by the Department of Health and Welfare of the Executive Yuan, Taiwan. Weights and heights of all participating children were measured using one and the same electronic meter. 

### 2.4. Dietary Statue

Food intake was surveyed via questionnaires completed by the parents and kindergarten teachers to document the subjects’ dietary status. The questionnaires were derived from the standardized questionnaire of the Nutrition and Health Survey, Taiwan (NAHSIT), which included frequency of food-intake, food-type, etc. [20]. Based on all 33 items categorized in the NAHSIT, four new items were added: fried foods, high-fat snacks, high-sugar snacks and sweetened beverages [15]. Subsequently, data from the parents’ questionnaires were pooled with those of the kindergartens regarding the subjects’ timing for food-intake daily (Cronbach α = 0.854). Accordingly, eight categories were used for the assessments, including three nutrient categories (i.e., calcium, protein & carbohydrates) and five food categories (i.e., vegetables and fruit, sweetened beverages, non-sweetened beverages, candy and fried food).

### 2.5. Psychomotor Development and the Chinese Child Development Inventory (CCDI) Scales

The CCDI (Chinese Child Development Inventory), modified by Hsu et al. post-adaptation from the Minnesota Child Development Inventory (MCDI), [21], has been frequently used to screen or assess children’s and infants’ physical and mental development [22,23]. As the MCDI having been amended for its application via the Chinese language, the CCDI derived has been shown to be reproducible through the calculated scatter-gram, curvilinear regression and the coefficient of correlation summed [21]. The scales of CCDI included a total of 320 items on the list over seven developmental areas (i.e., gross motor, fine motor, expressive language, comprehension-concept, situation comprehension, self-help, personal-social) and one summary scale (for the general development). Thus, it is suitable for the infant of 6-month to 6-yr old, including a questionnaire completed by the caregiver.

Moreover, validity of CCDI has been determined adequate due to the fact that its correlation coefficients for all of the developmental areas were above 0.90 and the general development was ~0.837 [21]. Based on other reports and to our knowledge, CCDI has also been shown useful to assess the children who may have development delays with strong reliability (up to 0.94) [22,23,24]. In our present analyses, the test-retest reliability of all test items employed was estimated ~0.934 and the mean score of differences between the pre-test and post-test was 1.61% (ranging from 0.62%~3.75%; data not shown); despite that it has been estimated that the correlation of CCDI with the draw a person (DAP) test is as good as 0.70 and that of the Chinese version of Denver Developmental Screening Test is higher than 0.70 [21]. 

### 2.6. Statistical Analyses

In present study, level of statistical significance was set at *p* < 0.05; where students t-test was to compare variables among the gender and various dmft scales (≤2 vs. ≥3, etc.), and one-way ANOVA was to compare variables among different age-groups. A multiple linear regression model was to analyze the relationship between all variables, where the cut-offs of dmft (as independent variable) and sex/gender, age groups, nutrition, BMI, father & mother education, father & mother jobs, oral habits as control variable were studied. All data sets were analyzed using SAS8.2 (SAS Institute; Cary, NC, USA). 

## 3. Results

A sum of the original 353 out of total 401 kindergarteners who completed the oral examinations and the surveys by parents/guardians were employed in our present study (i.e., 183 males: 51.84%, 170 females: 48.16%) and then divided into the groupings of 4-, 5- and 6-year olds (*n* = 140: 39.66%, 181: 51.27% & 32: 9.07%, respectively; see Table 1). Their mean-height was 108.88 ± 5.59 cm and mean-weight was 19.92 ± 3.03 kg. Based on BMI scores, there were 5.95% (*n* = 21) under-weighted, 62.61% (*n* = 221) normal, and 31.44% (*n* = 111) over-weighed children. The mean caries (dmft) score was 6.88 ± 5.17, and the mean of decayed, extracted and filled teeth was 5.78 ± 4.92, 0.09 ± 0.41, and 1.00 ± 1.97 respectively. Notably, the prevalence of caries in the present cross- sectional study was 85.27% (data not shown), which was significantly higher than that of the previous national survey described [25], and the rest of 14.73% present participants were caries-free (*n* = 52). However, the prevalence of caries did not differ between the boys and girls, similar to others’ report [26]. Further, there was no significant difference between the girls and boys, regarding dmft, height, weight, BMI or the distribution across all three BMI-categories. Analyses of the diets by food categories and the frequency of food intake showed that there was no significant difference between the boys and girls, either (Table 1). Despite the fact that certain CCDI measurements were variably spread out between the boys and girls, there were no statistically significant differences detected (Table 1). On the contrary, as the age increased, there were significant increases in the children’s dmft scores (*p* = 0.0016), height (*p* < 0.0001) and weight (*p* < 0.0001) measured over time, respectively. Further, regarding the developmental quotient (DQ) measured, their differences were rather statistically significant in all areas, except the gross motor and personal-social categories. This may suggest that there were sequential developmental changes or/and a deficiency as the function of time in the present kindergarteners analyzed.

In this study, it was hypothesized to reassess and confirm whether the higher levels of ECC is indeed associated the child’s psychomotor deficiency or not (i.e., via CCDI scales). To this end, we proposed to reassess by addressing the psychomotor status of the kindergarteners in the rural township of central Taiwan, where severity of ECC (e.g., dmft ~6.88) was much higher than that from the urban cities of southern Taiwan reported previously (~4.07) [15] and the national reports (3.67~4.35) [17]. Interestingly, their socio-economic status and the affordable access to healthcare facilities were rather comparable and close to those reported in other cities and areas (see Table 2), suggesting that these two common factors were not associated with the differential manifests of psychomotor deficiency identified in the present study [27]. Further analyses of the psychomotor development showed that the lower scores detected were reversely associated with higher dmft measures, albeit with few irregularities (see Table 3). Thus, the summed dmft scores were deemed as the representative of ECC (or alike), which was then computed with CCDI measures for the statistical comparisons. Furthermore, the dmft scores were stratified by sequential cut-offs individually (i.e., 2, 3, 4, etc.) to reveal any potential differences among the CCDI scales computed. When dmft = 6, 7 was separately employed as the cutoff, there was a significant difference in the general development measured (i.e., dmft ≤ 6 vs. ≥7; ≤7 vs. ≥8). Conversely, when dmft was ≥9, no significant difference was otherwise detected (in the general development: see online Appendix A, not shown here). Therefore, it is specially noted that: (i) CCDI scale was employed as the dependent variable, whereas the dmft, age, BMI and diet measures were independent variables, and (ii) though dmft measured did not constitute the normal distribution [28]; it is conceivable that the parametric methods, such as: the student t-test and ANOVA, were commonly applied [29], compared to the non-parametric analyses, which often produced some distortions of the original information and data. Nevertheless, employing a general linear regression for reassessing or remodeling dmft scores, despite having non-normalized distribution or potential under-estimation [28], has been generally applied [29]. Meanwhile, the same approach employed above has been used in several other analyses or models [30]; for example, the reported correlation between the socio-economics and dmft scores. 

It is though that dmft activity may be associated with child’s development over time. To explore the influence of any confounders on dmft measures during the child development, the Pearson correlation and partial correlation analyses among dmft and CCDI scale were applied, whose results showed that dmft measures were significantly correlated, though at much lower levels, with the development-related factors (i.e., age & gender; data not shown). Therefore, to address the relationship between dmft measures and CCDI vs. all other factors (i.e., age, gender, BMI, nutrition, parents’ educations & jobs, oral habits etc., as the control variables) were employed for the multiple regression analyses, as appropriate. In our regression analyses, CCDI scales were employed as the dependent variables and the relevant measures as independent variables, including caries (i.e., dmft = 2, 3, 4, respectively), age, gender and the consumption frequency of the food categories, oral habits (i.e., frequency of hygiene practice) as well as the parents’ education and vocation. These results are shown in Table 4, among the different regressions computed, the significant relationships were notably detected between the expressive language and the dmft scores of 2, and the comprehension-concept and the scores of 5–7 as well. Most of all, it is clearly demonstrated that when higher dmft activities were measured (i.e., from the rural township of central Taiwan), there were significant manifests in the psychomotor deficiency detected as well (see Table 5). Importantly, there was a similar trend detected in the present and the previous studies, as the higher dmft activity was positively associated with different psychomotor deficiencies via the bi-township comparison.

## 4. Discussion

The young children’s general growth involves both physical and mental development. Our present findings strongly suggest that higher levels of ECC (dmft > 3~8) in kindergarteners was indeed associated with specific manifests of developing psychomotor deficiency (i.e., expressive language and comprehension-concept). The mean dmft scores presently analyzed was 6.88 ± 5.17, significantly higher than that of the previous study reported (4.07) [15], the national reports (4.35) [25], European countries (2.0) [31] and USA (1.17) [26]. Interestingly, a recent result with similar trend on the psychomotor deficiency regarding personal-social and expressive language was identified in the 4-to-6-yr-old kindergarteners with “lower” dmft scores (~4.07) and “the lesser” psychomotor deficiency in a previous study [15] than those reported in our present cross-sectional study with a bi-township comparison in parallel (Table 5). Thus, it is clear that there was indeed a positive correlation, undiscovered previously, between severe ECC and specific psychomotor manifests (i.e., comprehension-concept, self-help and personal-social) in aged 4–6 children [15]. Based on these analyses and results, we have proposed a new hypothesis about severe caries in the growing children, if not properly treated, ECC or alike may not only cause loses in the dentition, but result in or induce the sequelae accompanied by developing psychomotor deficiency (i.e., personal language-communication to psycho-social interactions, etc.). It is worth noting that this detected relationship as being reassessed and confirmed at present may become a more generalizable phenomenon than what have been previously anticipated.

Further, in our bi-township analysis (Table 5) along with higher vs. lower caries detected, their socio-economic status and the accesses to oral health-care and facilities were comparable to those described in other areas/cities, suggesting that such manifests of the psychomotor deficiency cannot be solely attributed by the subjects’ characteristics involved, when they had rather comparable access to the national healthcare facilities vs. dental services, and their socioeconomic status (Table 2). Conceivably, severity of ECC may be independently associated with the psychomotor deficiency, not causally related to other external & environmental variables [15]. In parallel, we did recognize that dmft measured may not be summed to fully model the distribution on the normality, resulting in lopsidedness over the final outcomes or mechanisms being explored. Although we cannot completely rule out any potential contributions to psychomotor deficiency from other confounders (i.e., age or gender; data not shown), when the host’s ECC activity is high. Nevertheless, the present findings clearly demonstrated and confirmed that, based on the study design and the bi-township comparison where the parametric and linear-regression models were employed, it remains to be further investigated whether there are other un-identified or differential attributes (i.e., via direct neurophysiologic paths or indirectly outside or external to the neurologic circus; see Figure 1 and legend) existed between differential ECC activities and psychomotor deficiency that precludes to rectifying the hypothesis proposed [27,30] (Figure 1 for a new hypothesis). In addition, our hypothesis-driven exploration must further re-evaluate the weights of the kindergarteners with special needs, disabilities and/or systemic disorders (i.e., requiring a physician to screen all enrolled subjects) to balance the study outcomes on the CCDI scores in the future, despite the fact that there was no such cohort involved at present, as described and noted above. 

The questionnaire employed in MCDI [32] was completed by the parents and has been shown to have high validity [33]. For the toddlers at age 2, the language scales used in MCDI have been found significantly correlated with the predictive values of MLU (Mean Length of Utterance) and Sequenced Inventory of Communication Development, indicating that the expressive language scale of MCDI is a valid predictor on language development for the 2-yearr-old children [34]. Thus, along with the progression of CDI used in early childhood [35], MCDI has also recently been employed to estimate the levels of language development on skills with success [36]. In parallel, CCDI notably carries the same quality as that of MCDI [37]. CCDI and MCDI have been used in assessing the children’s general development [38], which demonstrated, as above, the good association with reasonable predictability among all tests analyzed (i.e., CCDI & BSID: 0.53~0.72; MCDI & BSID: 0.5~0.9) [38], including the comprehension-concept on language development or/and skills. Additionally, CCDI is regarded to carrying the same quality as MCDI, yielding valid indexes of identifying young children at risk for the delays and deficiency in early language development [37]. Nevertheless, these manifestations on the development may be associated with the central issue of our current findings, that the higher dmft scores, the less language development, potentially affecting both the receptive and expressive abilities on psychomotor measures for deficiencies in the growing children. 

Despite the fact that the underlying mechanisms remain unclear, there are several pathways that may explain the results obtained here, where severe ECC or alike can affect the child’s general, psychomotor or psychological developments, yielding oral pains or even tooth losses, reducing dietary intakes, loss of nutrients, weight, or sleep, etc., thereby leading to a poorer quality of life [3]. Furthermore, any confounding variables (i.e., age and gender) and co-modifiers (i.e., external behaviors or any other environmental measures on psychomotor risks. [27]) associated with the outcomes of CCDI and MCD over time must be held accountable for further examination in order to verify and rectify their potential impacts on the present findings (Table 4 and Table 5, Figure 1). Alternatively, severe ECC may lead to a deficiency or compromises in the mastication forces, verbal skills, and metabolism of the brain, etc., as proposed [7]. For the kindergarteners, such effect(s), when persistent in the long term, will likely change their personal interactions with the family and/or peers at school or in the community, all of which influence their motivations to social engagement and/or learning (i.e., language expression, verbal skills and communication, etc. [27]), thereafter triggering the related developmental deficiency or delays over time.

Moreover, based on the results of our previous reports [7,10] and the present study, it is conceivable that severe ECC or alike may have profound influences associated with development of psychomotor deficiency, as depicted in the summary of Table 5 and Figure 1**.** In addition, SLI may be associated with alternative conditions involving not only language-learning impairment but developmental deficiencies in language. With respect to how severe ECC may initially trigger their sound distortion [39], mis-articulation, or poorer oral functions, even malocclusion [40] leading to deficient language, maturation must be further testified before any definitive conclusion is drawn. For that, a large scale longitudinal prospective follow-up study has been launched to reveal this intriguing mechanism(s). Once this is achieved, the new insights and knowledge obtained will be implemented as an applicable tool or protocols, from the individual, community, and country, for caries prevention or an intervention program and specific policies for reducing the negative impacts of caries and improving the overall health effects for the children’s general development.

## 5. Conclusions

The results of our present cross-sectional study and the bi-township analysis indicate that there is indeed a positive relationship between the higher levels of ECC (i.e., dmft >3~8) and developing manifestations of psychomotor deficiency (i.e., expressive language & comprehension-concept scales) in the aged 4–6 kindergarteners analyzed. Further understanding of the underlying causal or major risk(s), once fully explored, will facilitate the establishment of better oral health-care protocols, prevention strategies, and polices to support the wellbeing and general development of young children in the future.

## Figures and Tables

**Figure 1 ijerph-16-03082-f001:**
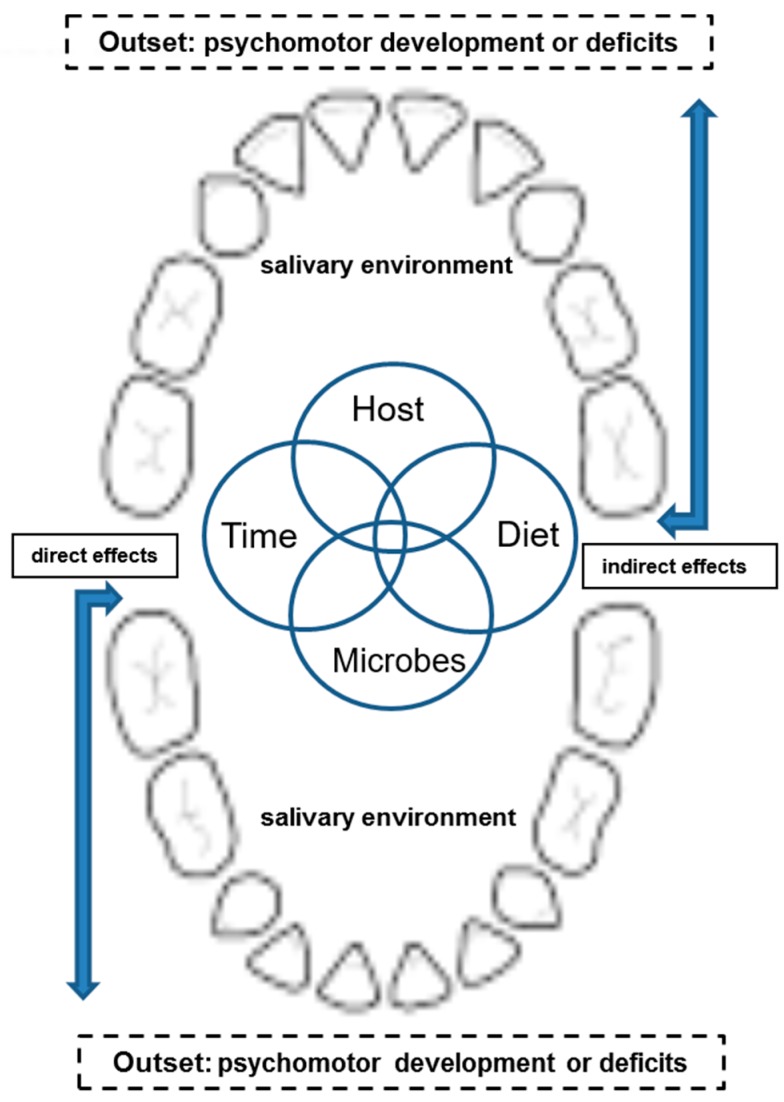
A new hypothesis is proposed, with edited modifications of the Classical Kele’s four circles during the initiation or progression of dental caries. Figure legend: A new hypothesis is proposed, based on the Classical Kele’s four Circles (i.e., host, diet, microbes, time), where the clinical manifests of psychomotor deficiency, in the context as an outset during the development, may interactively influence “directly” (i.e., via neuro-physiologic circus) or “indirectly” (i.e., via outside the neurologic circus) all 4 components on the initiation and/or progression of ECC in our oral cavity sounded by the saliva & salivary environment. Some of the external factors may include the knowledge, attitude and behaviors (i.e., hygiene habits), or any other environmental measures (i.e., disability, special needs, etc.) affecting the psychomotor risks [27].

**Table 1 ijerph-16-03082-t001:** Dental caries, CCDI, dietary status, height, weight and BMI of different gender and age analyzed.

(co)-Variables	Sex			Age (years)			
	Male	Female	*p*	4	5	6	*p*
	(*n* = 183)	(*n* = 170)		(*n* = 140)	(*n* = 181)	(*n* = 32)	
	(Mean ± SD)	(Mean ± SD)		(Mean ± SD)	(Mean ± SD)	(Mean ± SD)	
**Dental caries (decayed, missing and filled teeth, dmft index)**	6.84 ± 5.21	6.92 ± 5.15	0.8826	5.66 ± 4.67	7.66 ± 5.44	7.75 ± 4.85	0.0016 *
**Chinese Child Development Inventory (%)**							
Gross motor	81.08 ± 18.97	81.15 ± 20.11	0.9756	81.50 ± 16.50	81.51 ± 20.15	77.53 ± 25.24	0.5992
Fine motor	96.82 ± 19.02	96.50 ± 19.60	0.8862	104.08 ± 21.80	93.10 ± 16.55	89.36 ± 14.81	0.0000 *
Expressive language	90.98 ± 17.97	90.08 ± 19.77	0.6869	97.38 ± 19.51	86.89 ± 17.17	86.26 ± 18.13	0.0000 *
Comprehension-concept	88.20 ± 20.70	88.65 ± 20.92	0.1851	115.00 ± 25.89	105.05 ± 16.75	93.10 ± 13.04	0.0000 *
Situation comprehension	97.50 ± 14.37	98.06 ± 12.55	0.8527	93.92 ± 22.51	85.68 ± 18.51	83.01 ± 22.15	0.0023 *
Self-help	91.75 ± 18.12	89.97 ± 18.77	0.4114	96.70 ± 21.50	88.15 ± 15.02	84.94 ± 18.12	0.0002 *
Personal-social	86.22 ± 18.15	88.31 ± 1854	0.3369	89.42 ± 15.39	86.26 ± 19.29	83.87 ± 22.25	0.2448
General development scale	103.51 ± 15.20	100.06 ± 17.54	0.0727	110.05 ± 19.55	98.07 ± 11.96	93.20 ± 11.96	0.0000 *
**Dietary classifications (times/day)**							
Protein	4.52 ± 3.04	4.28 ±3.22	0.5218	4.53 ± 3.52	4.32 ± 2.83	4.46 ± 3.11	0.8794
Calcium	2.11 ± 1.65	1.83 ± 1.37	0.1382	2.22 ± 1.78	1.87 ± 1.38	1.76 ± 1.32	0.1561
Carbohydrates	4.47 ± 2.41	4.22 ± 3.48	0.4893	4.23 ± 2.41	4.53 ± 3.38	3.85 ± 1.83	0.5054
Vegetables and fruit	2.34 ± 1.82	2.34 ± 2.20	0.9836	2.27 ± 2.00	2.41 ± 2.06	2.25 ± 1.66	0.8460
Sweetened beverages	0.93 ± 0.96	0.92 ± 0.89	0.9091	0.97 ± 1.10	0.86 ± 0.71	1.09 ± 1.23	0.3987
Non-sweeten beverages	0.27 ± 0.48	0.29 ± 0.47	0.6634	0.29 ± 0.43	0.26 ± 0.48	0.33 ± 0.58	0.7461
Candy	3.14 ± 2.89	3.45 ± 3.14	0.4051	3.42 ± 3.36	3.14 ± 2.80	3.53 ± 2.74	0.7009
Fried foods	0.30 ± 0.35	0.37 ± 0.45	0.1520	0.35 ± 0.47	0.33 ± 0.36	0.30 ± 0.30	0.8543
**Height (cm)**	109.41 ± 5.48	108.32 ± 5.66	0.0675	106.11 ± 5.20	110.56 ± 4.96	111.57 ± 5.57	0.0000 *
**Weight (kg)**	19.88 ± 2.68	19.96 ± 3.38	0.7959	18.83 ± 2.57	20.45 ± 3.09	21.73 ± 3.03	0.0000 *
**Body mass index (BMI)**	17.25 ± 1.99	16.24 ± 1.68	0.0000 *	16.81 ± 1.91	16.75 ± 1.99	16.68 ± 1.43	0.9087
**BMI group**	*n* (%)	*n* (%)	0.0006 *	*n* (%)	*n* (%)	*n* (%)	0.0594
Underweight	11 (6.01)	10 (5.88)		10 (7.14)	11 (6.08)	0 (0.00)	
Normal weight	131 (71.58)	90 (52.94)		83 (59.29)	122 (67.40)	16 (50.00)	
Overweight	41 (22.40)	70 (41.18)		47 (33.57)	48 (26.52)	16 (50.00)	

*: *p* < 0.05.

**Table 2 ijerph-16-03082-t002:** The socio-economic status and health-care facilities for dental care-and-services in major cities of Taiwan analyzed.

City	The average of Disposable Income Per Household (NTD/USD Equivalent)	The Average of Disposable Income Per Person (NTD/USD Equivalent)	The Reported Numbers of Dental Services in the Country or Region (City/County)	The Averaged Numbers of Reported Dental Services Per Person in the Country or Region (City/County)
Taiwan	$891,249/$30,733	$254,643/$8781	27,984,750	1.23
Kaohsiung	$931,446/$32,119	$275,576/$9503	2,327,941	1.54
Tainan	$849,868/$29,306	$240,758/$8302	1,236,223	1.64
Nantou	$738,359/$25,461	$220,406/$7600	514,759	0.96

Legends: The official registry of the socio-economic status and the access to health-care facilities for dental care and services in Taiwan, and for the major cities reported and analyzed (i.e., cities of Kaohsiung, Tainan vs. the Nantou county). Note: The source of the above information was derived from the official health-care registry of Taiwan as below: http://ebas1.ebas.gov.tw/pxweb/Dialog/statfile9.asp; http://www.nhi.gov.tw/webdata/webdata.aspx?menu=17&menu_id=661&WD_ID=689&webdata_id=1399.

**Table 3 ijerph-16-03082-t003:** The mean scores of CCDI measured with different dmft scores analyzed.

The Mean Scores of CCDI	dmft (*n*)														
	0 (52)	1 (18)	2 (24)	3 (20)	4 (16)	5 (21)	6 (22)	7 (23)	8 (25)	9 (24)	10 (17)	11 (21)	12 (13)	13 (17)	14 (15)
Gross motor	72.66	86.50	85.07	76.17	85.15	79.93	84.00	83.45	81.26	89.03	76.70	73.84	68.08	85.15	85.94
Fine motor	98.03	101.69	103.06	95.50	100.64	93.84	104.65	98.92	90.68	99.66	91.60	91.84	82.57	101.19	94.49
Expressive language	90.95	100.60	97.05	87.89	92.64	86.66	95.06	90.14	85.61	94.87	86.62	85.33	82.18	90.90	94.90
Comprehension- concept	103.93	115.00	117.98	109.70	111.43	112.12	118.56	103.69	99.30	104.87	105.29	102.81	98.39	104.38	111.95
Situation comprehension	87.34	94.42	97.15	78.69	95.82	86.33	93.86	91.02	84.84	91.29	85.69	87.06	72.66	91.55	91.20
Self-help	91.30	98.23	97.82	88.92	95.86	86.50	93.33	91.32	87.42	92.74	91.54	90.41	84.02	88.22	93.17
Personal-social	87.86	91.57	95.27	88.99	79.64	84.52	86.13	87.27	82.69	90.99	83.41	87.06	82.74	91.98	88.17
General development scale	100.16	107.85	107.86	99.05	105.33	103.24	108.25	102.61	96.36	103.52	99.02	99.90	92.06	103.24	102.45

**Table 4 ijerph-16-03082-t004:** The regression model computed for their relationships detected between all variables and psychomotor developments.

Dependent Variable: Expressive Language				Dependent Variable: Comprehension-Concept			
**dmft class by 2 (≤2 vs. ≥3)**				**dmft class by 5 (≤5 vs. ≥6)**			
	**B**	**se**	***p***		**B**	**se**	***p***
(Constant)	122.90	16.28	0.0000	(Constant)	152.10	18.02	0.0000
dmft class	−6.88	2.94	0.0203	dmft class	−5.85	2.92	0.0464
Age group	−6.57	2.06	0.0017	Age group	−8.39	2.30	0.0004
Carbohydrates	1.82	0.66	0.0065	Calcium	2.59	1.08	0.0171
Fried foods	8.72	3.86	0.0251	Carbohydrates	2.04	0.74	0.0062
Child brush tooth after eat sweet	−5.91	2.32	0.0118				
				**dmft class by 6 (≤6 vs. ≥7)**			
					**B**	**se**	***p***
				(Constant)	152.43	17.88	0.0000
				dmft class	−7.57	2.87	0.0090
				Age group	−8.00	2.30	0.0006
				Calcium	2.56	1.07	0.0173
				Carbohydrates	2.02	0.73	0.0062
				**dmft class by 7(≤7 vs. ≥8)**			
				(Constant)	150.40	17.96	0.0000
				dmft class	−6.42	2.90	0.0281
				Age group	−8.08	2.32	0.0006
				Calcium	2.53	1.07	0.0195
				Carbohydrates	2.02	0.73	0.0064

(All variables): Sex, age group, nutrition, BMI, father & mother education, father & mother jobs, oral habits and the cut-off of dmft scores.

**Table 5 ijerph-16-03082-t005:** The cross-sectional bi-township analyses, for a parallel comparison, with the resulting statistically significant differences detected, **as depicted below.**

Area Studied	Urban Cities from the Southern Taiwan [15]	Rural Township from the Central Taiwan (the Present Study)		
**Sample Size**	433	353		
**Gender (M+F)**	M: 230 + F: 203	M: 183 + F: 170		
**Means of dmft scores (range: 0–20)**	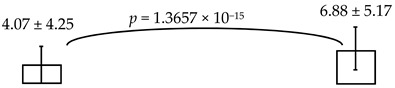			
**Expressive language (%)**	99.36 ± 12.93; 96.65 ± 11.91 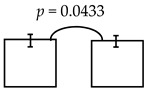	95.74 ± 16.59; 89.25 ± 19.10 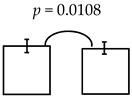		
	dmft ≤ 4 dmft ≥ 5	dmft≤2 dmft ≥ 3		
**Comprehension-concept (%)**		111.53 ± 20.75; 105.12 ± 21.28 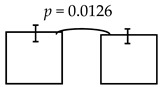	112.62 ± 20.29; 103.48 ± 21.23 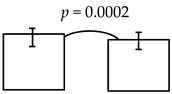	111.37 ± 21.74; 103.45 ± 20.07 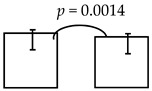
		dmft ≤ 5 dmft ≥ 6	dmft ≤ 6 dmft ≥ 7	dmft≤7 dmft ≥ 8
Personal-social (%)	89.23 ± 16.07; 85.17 ± 17.01 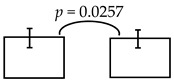			
	dmft ≤ 3 dmft ≥ 4			

Legends: The cross-sectional bi-township analysis was based on the demographics and the outcome variables compared between the two independent studies analyzed (e.g., from the urban cities of southern Taiwan vs. the rural township of central Taiwan).

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
