# Peer review of "Higher Levels of Early Childhood Caries (ECC) Is Associated with Developing Psychomotor Deficiency: The Cross- Sectional Bi-Township Analysis for The New Hypothesis"

_ijerph, 2019, doi:10.3390/ijerph16173082_

Round 1

Reviewer 1 Report

Dear Authors, 

congratulation for Your study. However there are some correction to be done before it is suitable for publication. 

1) Abstract has to be completeley rewritten in order to be more attractive to the reader

2) Introduction: please mention also the enamel structural defects as one of factors that favor caries deases (such as enamel development defects, mih 

https://www.ncbi.nlm.nih.gov/pubmed/28045323

https://www.ncbi.nlm.nih.gov/pubmed/30200640) 

"The present results indicated that there was indeed a positive correlation between severe ECC and different manifests of psychomotor deficiency (i.e., expressive language and comprehension-concept) in aged 4-6 children from the rural township of Nantou County in central Taiwan. These new findings, as the underlying risk(s) being explored [18] for a new hypothesis proposed herein, may contribute to our better understanding of its potential causes and/or risks, which will be applicable to the new health-care policies or novel strategies practical for the prevention of childhood dental diseases, such as ECC, in the future." rewrite without anticipating the results. 

3) Results The mean caries (dmft) score was 6.88±5.17, and the mean of decayed, extracted and filled teeth was 5.78±4.92, 0.09±0.41, and 1.00±1.97 respectively, compatible with the findings that we recently reported [13]. Notably, the prevalence of caries in the present study was 85.27% (data not shown), which was significantly higher than that of the previous national survey described [23] and the rest of 14.73% participants were caries-free (n=52). However, the prevalence of caries did not differ between the boys and girls, similar to others [24]. rewrite (also the following paragraph without commenting or comparing. it is indeed part of discussion

tables need a better description

Discussion: 

since there is this association between these children, You can discuss also to include the dental special care first visit as cumpolsory in the protocol to follow when a physician is facing this type of diabilities. 

discussion should be more clear. in some passages it is difficult to read and to uderstand. 

Reviewer 2 Report

the hypothesis requires more explanation and the external factors are not clear here , the discussion mentions these factors but is not reflected on the diagram  .

Reviewer 3 Report

The aim of the study is to reassess and confirm the relationship between severe childhood caries and the manifests of psychomotor deficiency in 4-to-6-yr-old preschoolers. The study is valuable to address the impact of oral health in children on their overall health and development. However, there are major issues with statistical analysis and study design in subject inclusion and exclusion criteria as following:

1.     Introduction section: need better organization to list existing study and findings, the gaps, current study aims and significance. Avoid putting current study results in the introduction section.

·      In the 2nd paragraph, it is important for the authors to describe how the previous study was conducted to give the conclusion of  “high levels of caries, or ECC alike, can result in oral-facial pain, reducing dietary intakes, loss of weights or/and sleeps, accompanied by the poorer quality of life” as well as connection of the ECC with visual, auditory etc. dysfunctions.

·      The author may use “reassess” in the study aim if the association or connection of the ECC with neurological or psychomotor deficiency is well established and by the same research of the groups of the previous study.  Otherwise, using “explore” or “assess” may be less misleading of the readers.

·      In the 3rd paragraph, 1st sentence, it is not clear whether the author is intended to list the fact of previous study or the hypothesis of the current study.

·      In the 3rd paragraph, please define how the two townships are “randomly” selected (the randomization method used to select the study location or population).

·      Please avoid putting current study results in the introduction section.

2.     Method and Materials.

·      1st paragraph should be re-organized to summarize the overall study design of the current study to give a clear map of how the current study was conducted. Cross-sectional study, longitudinal, etc.

·      Subject recruitment: again describe what randomization process and method was used to “randomly select the study location and population”.

·      Since it’s not a longitudinal study, rejection (denial) rate may be a better word than dropout rate. Additionally, there were no clear inclusion and exclusion criteria for the study. It is not clear whether subjects with developmental delay and systemic diseases etc. that can affect the subject’s psychomotor development were excluded from the study or not.

·      2.5. Psychomotor development and the Chinese Child Development Inventory (CCDI) scales: Please describe precisely how the CCDI scale is administrated, collected and analyzed. Explain how the estimated correlation coefficient was calculated. Please use scientific writing style to be precise in the methodology section.

·      Statistical analyses: There are major problems in statistic analysis. The age and gender can obvious confounding the status of dental caries and psychomotor development. Obviously, both caries severity and psychomotor development showed correlations with increasing age of the child. The current analysis proposal did not control or adjust the confounding factors such as age and gender and is likely leads to a false conclusion.

3.     Results/ discussion/conclusion

·      Better organization of the result section is needed.

·      The summary of the results was not clear on what statistical method is used and failed to control the confounding factors in the correlation between caries and psychomotor development. Therefore, the results/discussion/conclusion may not be valid and can be misleading by the confounding factor and not valid.

·      The authors included comparisons between the current study with the previous urban study. In this case, this should be included in the study aim and the brief summary of the previous study and how the comparison was carried out need to be added to the method section.

The study asked a very valid question with a lot of effort on data collection that can provide important evidence on the linkage and ECC with child psychomotor development. However, the authors need to work with a good statistician on an appropriate data analysis plan to draw valid conclusions.
